# Metastatic Multifocal Malignant Peripheral Nerve Sheath Tumour in the Cervicothoracic Spinal Cord of a Dog Initially Mimicking Meningomyelitis

**DOI:** 10.3390/vetsci10020170

**Published:** 2023-02-20

**Authors:** Javier Espinosa, María Ortega, Martí Pumarola, Eduardo Fraga, Laura Martín

**Affiliations:** 1Neurology and Neurosurgery Service of Centro Clínico Veterinario Indautxu, Alameda de San Mamés 38, 48010 Bilbao, Spain; 2Neurology and Neurosurgery Service, Pride Referrals, Derby DE248 HX, UK; 3Mouse and Comparative Pathology Unit, Department of Animal Medicine and Surgery, Veterinary Faculty, Networking Research Center on Bioengineering, Biomaterials and Nanomedicine (CIBER-BBN), Campus UAB, Universitat Autònoma de Barcelona, Bellaterra, 08193 Barcelona, Spain; 4Fraga Diagnostic España, S.L., Mijas Costa, 29649 Malaga, Spain

**Keywords:** meningomyelitis of unknown origin, neoplasia, cerebrospinal fluid, magnetic resonance, histopathology, metastasis, immunosuppression, peripheral nerve sheath tumor

## Abstract

**Simple Summary:**

Malignant peripheral nerve sheath tumors (MPNSTs) are aggressive and locally invasive neoplasia with the ability to metastasize. MPNSTs are usually associated with the spinal peripheral nerves, less common cranial nerves; are histologically anaplastic and highly aggressive with local connective tissue and muscle invasion; and most require immunocytochemical and ultrastructural verification. The aim of the present case report is to describe the clinical presentation and atypical magnetic resonance imaging (MRI) findings of a nine year old cross-breed with histologically confirmed multiple MPNSTs. The dog initially presented with a two week history of ambulatory tetraparesis. An MRI showed a diffuse intramedullary lesion extending from the C3 to T3 spinal cord segments and the cerebrospinal fluid (CSF) analysis showed marked lymphocytic pleocytosis; thus, Meningomyelitis of Unknown Origin (MUO) was presumptively diagnosed. The dog was treated with an immunosuppressive tapering plan of glucocorticoids and cyclosporine. Despite significant improvement and the stabilization of the clinical signs, the dog acutely deteriorated seven months after and the MRI was repeated. Interestingly, the follow-up MRI showed multiple well-defined intradural-extramedullary mass-like lesions, affecting several spinal nerve roots along the cervicothoracic spinal cord. The histopathological examination confirmed a final diagnosis of multiple MPNSTs, and a metastatic spreading was suspected due to the presence of round cells invading the subarachnoid space. According to the reviewed literature, this is the first reported case of simultaneous MPNSTs in the cervicothoracic spinal cord of a dog.

**Abstract:**

A nine year old cross-breed dog was presented with a two week history of ambulatory tetraparesis and proprioceptive ataxia affecting all four limbs. Meningomyelitis of Unknown Origin (MUO) was presumptively diagnosed based on the magnetic resonance imaging (MRI) findings and cerebrospinal fluid (CSF) analysis. The dog received a tapering dose of glucocorticoids and cyclosporine, showing significant improvement and the stabilization of the clinical signs for seven months. After this period, the dog showed an acute clinical deterioration and a follow-up MRI revealed new multiple lesions affecting different spinal nerve roots along the cervicothoracic spinal cord. Following euthanasia, a final diagnose of multiple malignant peripheral nerve sheath tumors (MPNSTs) was made based on the histopathological examination. MPNSTs can affect the cranial nerves, spinal nerves or the associated nerve roots at any location and can lead to secondary spinal cord compression. The aim of the present case report is to describe the clinical presentation and atypical MRI findings of a dog with histologically confirmed multiple MPNSTs. According to the reviewed literature, this is the first reported case of simultaneous MPNSTs in the cervicothoracic spinal cord of a dog.

## 1. Introduction

Peripheral nerve sheath tumors (PNSTs) are classified as either benign (schwannomas, neurofibromas and perineuriomas) or malignant (MPNSTs), principally based on their histological features and aggressive behavior [1,2].

MPNSTs are a relatively rare malignant tumor originating from the diversely differentiated cells that form the peripheral nerves (Schwann cell with an admixed fibroblastic cell type) [2]. In animals, MPNSTs have been reported in several domestic species, such as dogs [2], cats [3], cattle [4], horses [5], goats [6] and pigs [7], and even among wild species, such as a tiger [8] and a snake [9].

In dogs, MPNSTs have been reported in different areas, such as in the cutaneous tissue [10], liver [11], diaphragm [12], vagus nerve [13], tongue [14] or intraocular space arising from the ciliary nerve [15]. However, spinal nerves in the caudal cervical and cranial thoracic region and cranial nerves are most commonly affected [1]. The majority of canine schwannomas, which are a benign form of PNST of the nervous system, arise unilaterally and intradurally within the spinal nerve roots or extradurally in the brachial plexus, specifically in the paraspinal nerves of segments C6-T1 and, much less frequently, in the lumbosacral plexus [2]. Approximately 45% of the MPNSTs are located in the nerve roots, close to the spinal cord, being the most common site, in addition to the nerve roots or nerves of the brachial plexus [16].

According to the reviewed literature, multiple and simultaneous MPNSTs affecting the cervicothoracic spinal cord have not been reported in dogs. Thus, we report the clinical, MRI and histopathological findings of an uncommon presentation of MPNSTs.

## 2. Case Presentation

A nine year old, female, spayed, cross-breed dog (6.1 kg) was referred to the Neurology and Neurosurgery service of Centro Clínico Veterinario Indautxu due with a two week history of acute onset lethargy and slowly progressive ataxia and weakness affecting all four limbs. Mild intermittent discomfort was additionally reported when standing up or going downstairs. The dog was treated by the referring veterinarian with meloxicam at 0.1 mg/kg PO, SID (once daily), gabapentin at 10 mg/kg PO, TID (three times daily) and tramadol at 2 mg/kg PO, BID (twice daily) over ten days prior to presentation with no significant improvement. The dog had never displayed similar clinical signs before presentation and was reported to be a healthy dog otherwise, with no ongoing or previous diseases nor treatments. 

On presentation, the dog had an unremarkable physical examination. Neurological examination revealed ambulatory tetraparesis, mild to moderate proprioceptive ataxia affecting all four limbs and intermittent scuffing on the left thoracic limb. The postural reactions were absent on both pelvic limbs and were reduced on the thoracic limbs, the left forelimb being more affected. Spinal reflexes and cranial nerve examination did not reveal abnormalities. Caudal cervical hyperesthesia was noted. The neurological examination was consistent with a cervical myelopathy affecting the C1–C5 spinal cord segments. 

The main differential diagnoses, given the clinical history and neurologic deficits, included degenerative disease, such a disc herniation, neoplastic process (primary or secondary) and inflammatory/infectious diseases.

The complete blood count and serum biochemistry submitted by the referring veterinarian were within the normal limits. A low field MRI of the cervicothoracic region revealed an intramedullary lesion extending from C3 to T3, hyperintense on the T2-weighted (T2W) and isointense on the T1-weighted (T1W) sequences. This lesion was mildly lateralized to the left and affecting both the grey and white matter. Marked contrast enhancement was evident in the dorsal aspect of the spinal cord from C6 to C7. Meningeal contrast enhancement was also observed (Figure 1).

A lumbar cerebrospinal fluid (CSF) tap revealed a marked mononuclear pleocytosis (2200 WBC/μL) with lymphocytic predominance and increased proteins (Pandy test was positive). The CSF was submitted to an external laboratory in order to rule out infectious diseases (*Neospora caninum, Toxoplasma gondii, Bartonella* spp.,* Borrelia burgdorferisensu Lato* and *Cryptococcus neoformans*) and lymphoma by means of a Polymerase Chain Reaction for Antigen Receptor Rearrangements (PARR); all of the results were negative. Based on the results of the diagnostic tests, an inflammatory condition was considered most likely; hence, a MUO was presumptively diagnosed. A neoplastic condition was considered less likely at that stage.

The dog initially received prednisolone 1 mg/kg PO SID and clindamycin 11 mg/kg PO BID while awaiting final CSF analysis. When the results came back, the clindamycin was stopped and the prednisolone increased up to 2 m/kg PO SID for two weeks, in which the dog showed a significant improvement. The dog was noted to be almost completely back to normal with just a very subtle proprioceptive ataxia affecting all four limbs and no other remaining deficits. Cyclosporine at 10 mg/kg PO SID was introduced at that time. Prednisolone was tapered by half dose every 4–6 weeks until reaching a dose of 0.5 mg/kg PO every other day, which was eventually maintained. The dog remained stable for the seven months following diagnosis. However, the dog showed an acute deterioration at that time. Neurological examination revealed moderate ambulatory tetraparesis, proprioceptive ataxia affecting all four limbs and postural reaction deficits in all four limbs, with the left thoracic limb being the most affected. The withdrawal reflex was reduced on the right thoracic limb and absent in the left thoracic limb. Cervical hyperesthesia was also evident along all the cervical spines. The second neurological examination was consistent with an extensive/multifocal lesion affecting the C1-T2 spinal cord segments.

A follow-up MRI showed multiple, well-defined, intradural-extramedullary mass-like lesions affecting multiple spinal nerve roots along the cervicothoracic spinal cord. These lesions were T2W hyperintense and T1W isointense compared with the normal spinal cord parenchyma and showed a marked and homogeneous contrast enhancement. The largest mass (1.1 cm × 0.8 cm) was located at the level of the left C1–C2 intervertebral foramen and on the region of the left nerve root of C1, causing severe compression of the spinal cord. The remaining lesions had similar features but were only mildly compressing the spinal cord. The previously seen ill-defined intramedullary lesion T2W hyperintensity was less severe and extensive than on the first MRI, but was then more severe and visible at the level of C3, which was not visible on the first MRI scan. The adjacent hypaxial and epaxial muscles and the left supraspinatus and subscapularis muscles were moderately reduced in size with multifocal T2W hyperintense areas most consistent with denervation atrophy and secondary myosteatosis (Figure 2).

Based on the follow-up MRI findings, a multifocal infectious/inflammatory disease or, less likely, an infiltrative neoplastic disease were considered the main differential diagnoses. To repeat, a CSF tap was discussed but eventually rejected by the owners on that occasion.

Given the poor prognosis and attending the owners’ will, the dog was euthanized. The cervical spinal cord, including the nerve roots, was removed and fixed in 10% neutral buffered formalin. 

The histological study revealed the presence of a dense neoplastic cell population infiltrating the extradural and intradural parts of the dorsal and ventral nerve roots. The majority of the neoplastic cells were medium or large in size, showing a spindle-shaped morphology with a prominent hyperchromatic nucleus, and marked anisokariosis was observed. The cytoplasm was scarce and pale and the cell boundaries were ill-defined. Rare mitoses were seen. A high number of pycnotic cells and necrotic foci were present. There were also some areas with a predominance of a different cellular pattern, and the neoplastic cells were smaller and rounded, with a pyknotic nucleus and a pale cytoplasm. Inflammatory lymphocytic cells were also present. All of the neoplastic cells were characterized by solid and cord-like growth patterns infiltrating the endoneurium, compressing and destroying the myelinated nerve fibres fascicles and ganglionar neuronal bodies, invading the perineurium and the dura mater and reaching the subarachnoid space. In the area of contact with the spinal cord, the nervous parenchyma suffered a marked compression, showing a diffuse spongiosis of the white matter, the presence of numerous spheroids and a glial proliferation. Multifocal perivascular cuffs of neoplastic cells were also present (Figure 3). The immunohistochemical (IHC) study revealed a high number of immune positive neoplastic cells, showing a strong dark staining against Sox10 (20–80%) and Laminin (60–80%). Other markers, such as S-100 protein and Vimentin, resulted in a low number (10%) of neoplastic immune positive, cells showing weak and pale staining.

The histopathology of the whole cervical spinal cord confirmed the presence of multiple MPNSTs bilaterally and dorsoventrally affecting multiple spinal nerves roots.

## 3. Discussion

The case described here initially presented with a two week history of acute onset ambulatory tetraparesis and incoordination in all four limbs. The anatomical localisation was the C1–C5 spinal cord segments, albeit the MRI showed an extensive lesion extending from C3-T3. The first MRI findings were consistent with an inflammatory/infectious process. Whilst they are non-specific, the commonly reported MRI findings in dogs with meningomyelitis of unknown origin are ill-defined, focal or multifocal, diffuse T2W hyperintense and T1W isointense intramedullary lesions, with variable degrees of parenchymal and/or meningeal enhancement after contrast administration. The T2W hyperintense signal can affect both the grey and white matter [17]. All of these features were present in the first MRI. However, spinal cord neoplasia can be indistinguishable from meningomyelitis on MRI. For instance, lymphoma MRI features can be very heterogeneous, including all the above-mentioned findings for meningomyelitis and, hence, lymphoma should be included on the differential diagnosis list when diffuse or multifocal lesions are detected in MRI. 

The CSF analysis showed a marked lymphocytic pleocytosis, and although further studies are needed, it is unusual to find CSF pleocytosis in dogs with spinal cord tumours [18]. Nevertheless, lymphoma was still considered as a differential diagnosis and, hence, a PARR test was performed, being inconsistent with lymphoma. The presence of lymphoblasts on the CSF examination is a key point on the definitive diagnosis of central nervous system (CNS) lymphoma, yet is a low-sensitivity test as lymphoblasts will only be identified in 30% of humans with CNS lymphoma [19]. In our case, no lymphoblasts or any neoplastic cells were identified on the CSF analysis in house or on the external laboratory examination. In addition, the predominantly lymphocytic pleocytosis is usually consistent with an immune-mediated condition. 

In general, the inflammatory processes affecting the CNS will provoke mild to marked CSF pleocytosis, whereas neoplastic processes usually produce none or mild pleocytosis, yet marked pleocytosis can seldom be seen [20]. In one study with 21 cases of presumptive spinal-only meningoencephalitis, the median total nucleated cell count was 209 cells/mm^3^ (ranging between 6 and 6000) [21]. 

In our case, the CSF was considered to be suggestive of an inflammatory process overall and, at that stage, any neoplastic condition was considered unlikely. Based on the first MRI and CSF findings together, our differential diagnosis included MUO and, less likely, a neoplastic process. The chosen immunosuppressive treatment was started, and the dog displayed a satisfactory response with an initial rapid improvement. Despite tapering the steroids dose gradually, the patient did not show any deterioration. Based on our patient’s CSF analysis, and combined with the clinical sings, MRI findings and the favourable response to immunosuppressive therapy, a presumptive diagnosis of MUO was confirmed.

However, seven months later, while on chronic cyclosporine (10mg/kg/day) and a low dose of prednisolone (0.5 mg/kg/every other day), the dog acutely deteriorated and a relapse of the MUO was suspected. The neurological examination was consistent with a more severe cervical lesion extending caudally and the second MRI confirmed a worsening of the previous condition or a new disease, with multiple lesions affecting several spinal nerve roots along the cervicothoracic spinal cord. Such lesions were not present on the previous study. Based on the follow up MRI findings, our differential diagnosis included multifocal granulomatous neuritis and, less likely, a diffuse neoplastic infiltration, such a lymphoma or metastatic disease. Owing to the multifocal nature of the lesions, a complete surgical resection was not technically feasible, and the owners rejected decompressive surgery and the biopsy of the most compressive lesion at the level of C1–C2. In hindsight, and if a pre-mortem final diagnosis would have been reached by means of a biopsy, stereotactic radiotherapy could have been considered an option as it has been shown that its application on the proximal portion of nerve roots seems an adequate treatment alternative to surgery in dogs with MPNSTs [22]. In the same scenario of hypothetical pre-mortem diagnosis, chemotherapy would not have been considered as MPNSTs are considered highly resistant to chemotherapy [23]. Based on the patient´s deterioration and attending the owners’ will, the dog was euthanized without complementary tests, such as a CSF tap. 

A definitive diagnosis of MPNST was only achieved through histopathological examination. In retrospect, a potential correlation between the marked lymphocytic pleocytosis detected in our case and a potential MPNST metastatic process remains unclear. In the present case, inflammatory lymphocytic cells were present within the neoplasia. Any neoplasm could be associated with a certain degree of inflammatory response, which would justify the presence of lymphocytes in this case. Moreover, in human medicine, the presence of tumor-infiltrating lymphocytes in MPNST is relatively frequent and some studies have found positive associations between the presence of lymphocytes and clinical response rates [24]. In the human and veterinary medicine literature, information regarding metastatic MPNST and marked pleocytosis is lacking. Interestingly, normal CSF results are reported in cases with intradural extension and confirmed CNS metastasis of MPNST in human medicine [25]. On a different note, Meningeal Carcinomatosis (MC) is defined as the multifocal or diffuse leptomeningeal infiltration of neoplastic cells, which can be seen with solid tumours affecting the CNS, such as sarcomas. Mild lymphocytic pleocytosis without neoplastic cells detection in the CSF associated with this condition is reported in veterinary medicine [26]. In our case, no neoplastic cells were found in the CSF, although further investigations to improve the sensitivity of CSF cytology, such as monoclonal antibodies techniques or tumours-specific markers [26,27], were not pursued. In any case, the final diagnosis of MC remained upon histopathological examination by identifying the neoplastic cells infiltrating the leptomeninges [26]. Neoplastic cells invading the subarachnoid space were found in our patient seven months after the CSF analysis.

MPNSTs are aggressive and locally invasive neoplasia. In human medicine, MPNSTs are reported to be an uncommon neoplasia affecting 1 in every 100,000 inhabitants, and up to 50% of these tumours are reported to occur in people with Neurofibromatosis type 1 (NF1), which is known to be a predisposing condition [28,29]. 

NF1 is an autosomal dominant condition characterized by the growth of benign tumours along the nerves of the skin, brain and other tissues. Some people with NF1 develop malignant tumours that grow along the nerves in adolescence or adulthood. Neurofibromatosis has also been reported in animals, such as cattle and dogs, causing multiple neural tumours that affect the nerves, viscera and skin, but no relationship with MPNSTs has been described in veterinary patients [30,31,32]. 

In dogs, MPNSTs tend to be present in the paraspinal peripheral nerves and they are the second most frequent spinal nerve tumour, after schwannomas. MPNSTs can also have a spreading growth and metastases outside the epineurium [2].

Multifocal MPNSTs affecting the CNS have been reported in humans, although it is a very rare condition. In the veterinary literature, a recent study describes four cases of an aggressive melanotic variant of MPNST affecting young dogs. In three of the four cases, the tumours were present multifocally, affecting the myelencephalon and cervical spinal cord in two cases and the lumbar spinal cord in one case [33]. Furthermore, concomitant MPNSTs and benign cutaneous PNSTs have been also reported in a dog [30]. In the present case, an independent growth of neoplastic cells of MPNSTs was considered less likely due to the wide presence of round neoplastic cells in the subarachnoid space.

In humans, up to seven of every ten patients with MPNSTs will develop metastatic spreading [23], most likely to the lungs, followed by bone and finally pleura. Three types of CNS metastatic pathways of MPNSTs are described: direct invasion, CSF spreading and hematogenous metastases [1,34]. Spinal cord and brain metastasis are very rare in non-neurofibromatosis Type 1 MPNSTs [35,36,37,38,39,40]. In the veterinary literature, only one case report describes a CNS metastasis of an intradural MPNST in a dog. This dog suffered a recurrence of the initial resected MPNST at the C2 nerve and metastasized at the cranial thoracic level [1]. Information of the MRI features of MPNSTs metastasis in veterinary medicine is scarce. The radiological description of human MPNSTs spinal metastases from sporadic case reports are mainly intradural contrast-enhancing lesions [1,35] or multiple oedematous infiltrations of the spinal cord in T2W images, showing diffuse hyperintensities and multiple enhancing point lesions in T1W images [1,41]. Based on the latter, one strongly considered possibility was that the diffuse intramedullary lesions detected on the first MRI were already a metastasis of a by-then non-visible MPNST.

Once again, the histopathological examination was essential to establish a definitive diagnosis. The post-mortem examination confirmed the presence of multiple MPNSTs bilaterally and dorsoventrally affecting multiple spinal nerves roots. The majority of MPNSTs will present several malignant cytological features, which were also seen in our case´s multiple masses. Such reported features [2] include the dense neoplastic hypercellularity of pleomorphic fusiform cells with a hyperchromatic nucleus showing anisokariosis, which in this case, were infiltrating the extradural and intradural component of the dorsal and ventral nerve roots bilaterally. A high number of necrosis foci is also reported [2] and were seen in our case, together with rare mitotic figures. IHC is often required for diagnostic confirmation of PNSTs and differentiation from other spindle cell tumors. A specific immunomarker for the definitive diagnosis of a MPNST and its differentiation from other nerve tumors or other spindle cell tumors is still lacking in veterinary medicine, and case-by-case or interspecies differences in IHC expression can occur, even when applying a broad diagnostic IHC panel [2]. Sox10 is a transcription factor that is crucial for the specification, maturation and maintenance of Schwann cells and melanocytes [42]. In animals, Sox-10 IHC has been utilized to differentiate PNSTs from perivascular wall tumors [2]. In both dogs [43] and humans [44], Sox10 is less expressed in MPNSTs than in benign PNSTs, and its expression also decreases in higher-grade MPNSTs compared with low-grade tumors. This trend most likely reflects the lower degree of Schwann cell differentiation in more malignant tumors. In the present case, only a subset of neoplastic cells exhibited immunolabeling with Sox-10 and S100, and this may reflect the downregulation of Schwann cell markers with the progression toward malignancy. These immunomarkers are more strongly and diffusely expressed in human schwannoma and are considered specific, but not sensitive, immunomarkers in the diagnosis of MPNSTs [45]. Vimentin, a major constituent of the intermediate filament family of proteins, is ubiquitously expressed in normal mesenchymal cells. Increased vimentin expression has been reported in various epithelial cancers, including prostate cancer, gastrointestinal tumors, nervous tumors such as nerve sheath tumors, breast cancer and lung cancer, among others. In veterinary medicine, due to its ubiquity, vimentin is used primarily to discriminate between epithelial and mesenchymal tumors and as an indicator of malignancy [46]. Given the multiple subtypes and variable expression of many IHC markers, MPNSTs can be difficult to differentiate from other soft tissue sarcomas, requiring extensive investigation with IHC, which may be cost-prohibitive in a diagnostic setting [10]. Taking into account the histopathological features of the present case, the differential diagnoses included other spindle cell-like sarcomas, such as schwannoma and neurofibroma, leiomyosarcoma, histiocytic sarcoma and a new entity, termed perivascular wall tumor. IHC study allows the differentiation of the above-mentioned neoplasia and MPNST [2]. When taken as a whole, our IHC results were more consistent with a MPNST than with other spindle cell sarcomas. 

MPNSTs lack an epineural tumour capsule and, hence, they can fiercely invade the surrounding structures. All neoplastic cells, in our case, had such an aggressive behavior, as they were infiltrating, compressing, destroying and invading the adjacent structures. In the area of contact with the spinal cord, the nervous parenchyma suffered a marked compression, as well as diffuse spongiosis of the white matter.

The fact of finding round cells in the subarachnoid space (Figure 3A) leads us to hypothesize that the case described here developed a multifocal occurrence of MPNSTs due to the metastatic spreading of a non-visualized MPNST on the first MRI.

As an important note, our patient was immunocompromised for several months due to prednisolone and cyclosporine treatment. Interestingly, in human medicine, there is one case report describing multifocal spinal MPNSTs in an immunocompromised patient, where accelerated tumour growth and early metastasis secondary to immunosuppression was considered [47]. There is also one case report in a dog with multicentric lymphoma that discusses the possibility of it being secondary to cyclosporine treatment, although this could not be proven [48]. In humans, cyclosporine-A associated malignancy is well known [49] and, hence, this immunosuppressive drug could also have played a role in this case. Immunosuppressive treatment could have been detrimental in our patient´s clinical progression, promoting the fast spreading and aggressive growing rate of a single MPNST, leading to acute and severe deterioration. This outcome is not usually reported in dogs with diagnosed MPNSTs, as they do not usually undergo an aggressive immunosuppressive treatment, which in our case, was initially started to treat a suspected MUO.

To our knowledge, our case is the first reported case of MPNSTs simultaneously affecting several cervical spinal nerve roots, which was putatively due to metastatic growth.

As a limitation of this study, a complete post-mortem examination or full body advanced imaging were not performed; thus, we cannot know whether systemic hematogenous metastases were present. However, the follow-up thoracic radiographs and abdominal ultrasound did not suggest a metastatic disease in the other organs. In addition, performing a CSF after the second MRI could have added valuable information to compare with the previous CSF analysis and to better characterize the condition. Further studies concerning MPNSTs CNS metastasis and the associated CSF features are needed, as well as studies of the potential link between immunosuppressive treatment and MPNST metastasis in veterinary medicine. 

## 4. Conclusions

Multifocal simultaneous MPNSTs involving multiple nerve roots of the cervicothoracic spinal cord have not previously been reported in dogs. This multifocal occurrence of MPNST was considered to be due to metastatic growth, probably caused by CSF. According to the reviewed literature, this is the first report that describes the clinical, advanced imaging and histopathological findings of multiple MPNSTs affecting the cervicothoracic spinal cord in a dog. The metastatic infiltration of the spinal cord and nerve roots should be included in the differential diagnosis when T2W hyperintense contrast enhancing intramedullary lesions are visualized on MR, especially in regard to MPNSTs and lymphoma.

## Figures and Tables

**Figure 1 vetsci-10-00170-f001:**
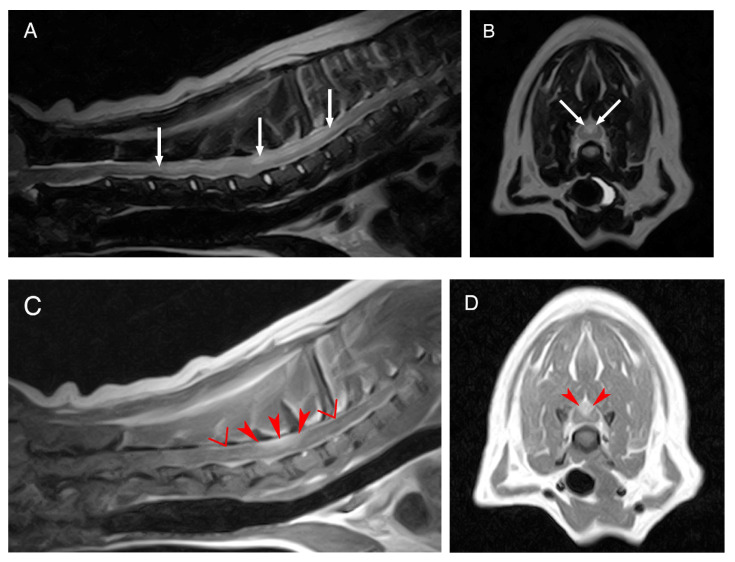
MRI of the dog’s cervical region. Midsagittal T2W image (**A**), transverse T2W image at the level of C6–C7 (**B**), mid sagittal T1W post contrast image (**C**) and transverse T1W post contrast image also at the level of C6–C7 (**D**). The white arrows are pointing to the T2-weighted hyperintense, intramedullary lesion extending from C3 to T3. The red arrowheads show the marked contrast enhancement in the dorsal portion of the spinal cord at the level of C6–C7. The red inverted triangles show the meningeal contrast enhancement.

**Figure 2 vetsci-10-00170-f002:**
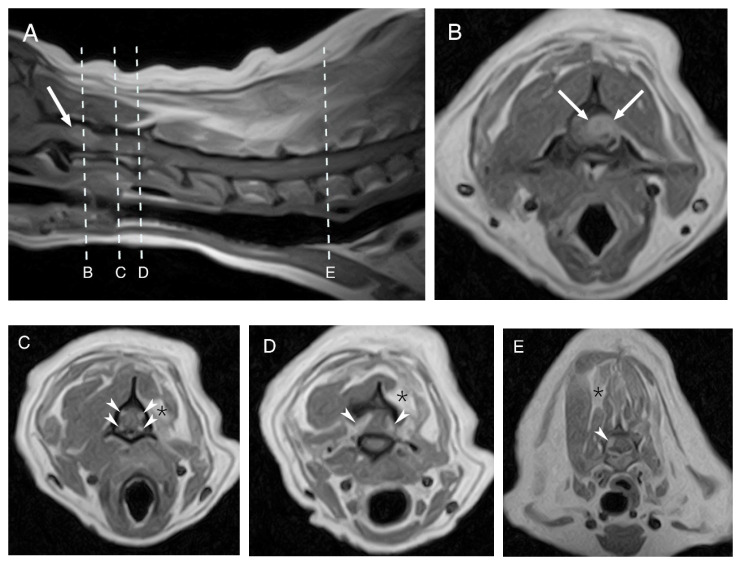
Follow-up MRI of the dog’s cervical region. Midsagittal T1W post contrast image (**A**) and T1W post contrast transverse images at different levels (**B**–**E**). The arrows are pointing to the largest mass at the level of C1–C2 (**A**,**B**). The arrowheads are showing the multiple lesions affecting several ventral and dorsal nerve roots bilaterally at the level of C2 (**C**), C2–C3 (**D**) and C6 (**E**). The asterisks show the secondary muscles changes at the level of C2 (**C**), C2–C3 (**D**) and C6 (**E**).

**Figure 3 vetsci-10-00170-f003:**
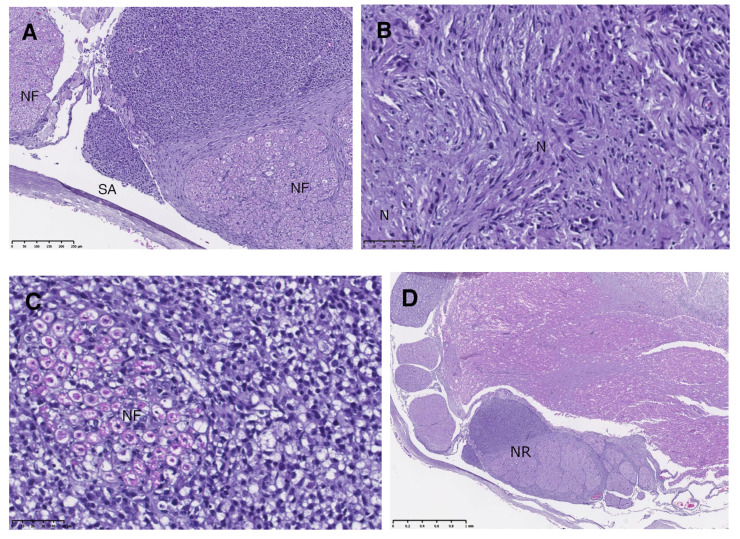
Histological features of the neoplastic lesion confirmed the presence of multiple MPNSTs (Hematoxylin and Eosin, H&E). (**A**) Nerve root invaded by neoplastic cells reaching the subarachnoid space (SA). Nerve fascicles (NF) are also visible. (**B**) Spindle-shaped neoplastic cells (N) population organized in bundles or irregularly distributed. (**C**) Small rounded neoplastic cells growing in a solid pattern and infiltrating nerve fascicle (NF). (**D**) dorsal and ventral roots (NR) thickened and deformed by the neoplastic population, compressing the spinal cord parenchyma.

## Data Availability

Not applicable.

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
