# Peer review of "Metastatic Multifocal Malignant Peripheral Nerve Sheath Tumour in the Cervicothoracic Spinal Cord of a Dog Initially Mimicking Meningomyelitis"

_vetsci, 2023, doi:10.3390/vetsci10020170_

Round 1
Reviewer 1 Report
The manuscript focuses an hughly interesting topic, which have been
explored in veterinary cancer research. The paper is in my opinion is correctly written and analizes all the main aspects of the pathology, therefore it is perfectly publishable.
I have only found some small discrepancies in the bibliography that should be rectified, since reference 1 and 42 are empty, and 41 does not appear in the text.
Reviewer 2 Report
Dear Editors and Authors, this case report describing a very rare condition is well written and well documented (describing the clinical signs, diagnostic protocols, findings, therapy, follow-up, and histologic examination) and deserves publication. Anyway, before publication, I have some comments regarding, in particular, the case discussion.
The discussion needs to be rephrased in some points and needs to be more consequential. I suggest starting with the clinical case summary, addressing and discussing all the critical points (MRI and CSF) that lead to "the wrong" diagnosis, and at the end, talking about MPNST.
Simple summary line 22: Please correct the term AMR. What does it mean? Is it an error or an acronymous?
Discussion line 187: please add: on the first MR and the CSF findings.."our differential diagnosis included..."
Discussion lines 191-194: please consider to move this part from this point to line 250
Discussion line 218: The authors discussed the CSF findings concerning MUO and Lymphoma deeply, but no discussion of the possible reasons for such a marked lymphocytic pleocytosis in this clinical case and related to the final diagnosis of MPNST. Please address this point because it is crucial. This marked pleocytosis is one of the reasons (and I agree with that) for considering a MUO the first differential in both RM studies. Moreover, the therapy targeted an immune mediated disease due to the imaging findings, the CSF findings, and the first differential.
Discussion lines 230-235: These sentences aren't clear . The dog was euthanized before reaching the exact diagnosis, so for what kind of disease radiotherapy is an option? Regarding the bibliography cited by the authors, I suppose that they referred to MUO. For the reader could be confounding if this is not specified. Then they talk about chemotherapy for MPNST but at this stage the didn't know the exact diagnosis and a MPNST wasn't a differential, so I don't see the reason for discussing this point. Please rephrase these sentences or consider to remove this part.
Reviewer 3 Report
Please see enclosed file.

Reviewer 4 Report
As written I am unconvinced that this is a neoplastic and not an inflammatory disease. There are a number of immune-mediated diseases that can cause inflammation of the nerve roots of dogs which could result in proliferation of spindle-shaped cells. This possibility is supported by the presence of lymphocytes as reported in the enlarged nerve roots. Such immune-mediated diseases are not uncommon and typically affect multiple areas. Furthermore, the history fits much better with an immune-mediated disease (sudden onset, response to anti-inflammatories for 7 months). The authors must provide convincing evidence why this is a neoplastic disease. Additional testing such as IHC could be useful. It is key for publication that the authors prove their diagnosis in this case.
I agree with the statement on Line 18 that immunohistochemistry is required for definitive diagnosis of a NST. However, IHC was not done in this case. If the authors can prove this is a neoplastic disease, they also have to provide conclusive evidence that this is a NST. This requires the use of immunohistochemistry and potential diagnoses should be discussed and excluded based on the IHC findings. This is particularly important in the case in which no full post-mortem was performed. It cannot be excluded that the neoplastic cells observed originated from a different location and have metastasized to the nerve roots.
Lines 214-229 are just a repeat of the clinical information and so can be removed.
The term ‘multifocal’ in the title and throughout is a bit misleading. This to me suggests that you have multiple neoplasms developing independently which you cannot prove in this case. Instead I would call this a “NST involving multiple nerve roots” instead.
Round 2
Reviewer 2 Report
Dear Editor, dear authors,
The authors have implemented the majority of the requested changes and the paper has improved substantially.
I have no more comments, and I suggest publication
Reviewer 4 Report
Thank you for including the immunohistochemistry. However, the results of the immunohistochemistry should be discussed. Most readers will have no idea what these antibodies are and why these results support your diagnosis. For example, I would have thought a PNST should have vimentin immunostaining. Please also include references for the validation of the antibodies used in dogs - just because something works and is specific in humans, doesn't mean it is similar in dogs. Likewise, the presence of lymphocytes within the neoplasm should be discussed as these are not listed as a feature of these tumors.
Overall, I feel that the manuscript lacks a good discussion on why you are so sure of the diagnosis. Just because a pathologist says it is this diagnosis does not make it necessarily correct. Include a paragraph that goes through potential differential diagnosis and really convinces the reader that the diagnosis that the pathologist has made really is the correct diagnosis.
I would also change this to a 'putative PNST' in the title and elsewhere. Without doing a full necropsy you will never be 100% certain of this diagnosis.
